# Shaking Table Test Research on the Influence of Center-Hung Scoreboard on Natural Vibration Characteristics and Seismic Response of Suspen-Dome Structures

**Suduo Xue [1], Zetao Zhao [1], Xiongyan Li [1],\*, Renjie Liu [2], Majid Dezhkam [1], Zhen Lu [1], Tifeng Liu [3], Qinxin Fan [4] and Hui Jing [4]**

1   Department of Architecture and Civil Engineering, Beijing University of Technology, Beijing 100124, China
2   Department of Civil Engineering, Yantai University, Yantai 264005, China
3   China MCC17 Group Co., Ltd., Maanshan 243000, China
4   Department of Architectural Design and Research, China Aviation Planning and Design Institute Co., Ltd., Beijing 100120, China
\*   Correspondence: xiongy2006@126.com

**Abstract:** The center-hung scoreboard is a large electronic display device, that usually has a significant weight and is flexibly suspended from the roof of the gymnasium. The suspen-dome structure is one of the common roof structures of gymnasiums. In order to investigate the effect of the center-hung scoreboard on the natural vibration characteristics and seismic response, including acceleration, displacement, and strain, of the suspen-dome structure and considering that the dynamic behavior of the structure under seismic action can be realistically reflected, a shaking table test was carried out based on the suspen-dome structure of the Gymnasium of the Lanzhou Olympic Sports Center. Firstly, according to the size of the shaking table, a dynamic scale model with a geometric similarity ratio of 1:20 was established. After that, the white noise excitation test and numerical modal analysis were conducted on the models with and without the center-hung scoreboard to compare the two modes' natural vibration characteristics. Furthermore, the earthquake simulation test was carried out on the models with and without a center-hung scoreboard, and their various seismic responses, including acceleration, displacement, and strain, were compared and analyzed. Finally, the acceleration response and displacement response of the center-hung scoreboard were investigated. The results show that the higher-order natural frequencies of the suspen-dome structure will increase when the center-hung scoreboard is suspended from the roof, and the swing of the center-hung scoreboard will be excited first in the low-order mode. In addition, the various seismic responses, including acceleration, displacement, and strain, of the model with the center-hung scoreboard are all increased compared to the model without the center-hung scoreboard. Meanwhile, the influence of the center-hung scoreboard on the seismic response of the suspen-dome structure decreases from the inner ring to the outer ring of the structure. Moreover, under the action of an earthquake, the acceleration response and displacement response of the center-hung scoreboard are both extremely high. Considering the center-hung scoreboard in the analysis and design stage of the suspen-dome structure is necessary.

**Keywords:** suspen-dome structure; shaking table test; center-hung scoreboard; natural vibration characteristic; seismic response

## 1. Introduction

### 1.1. Literature Review

The suspen-dome structure is a long-span prestressed spatial structure, that was first proposed by Kawaguchi in Japan [1,2]. The suspen-dome structure is mainly composed of a single-layer reticulated shell and a prestressed cable-strut system, as shown in Figure 1a. Figure 1b is the sectional view of the structure, which reflects its principal features; the

upper reticulated shell has high rigidity and directly bears the roof load, while the lower cable-strut system mainly contains hoop cables, radial cables, and vertical struts, which can improve the rigidity and strength of the overall structure, so that the suspen-dome structure has superior integrity and stability and can be applied to larger-span buildings [3].

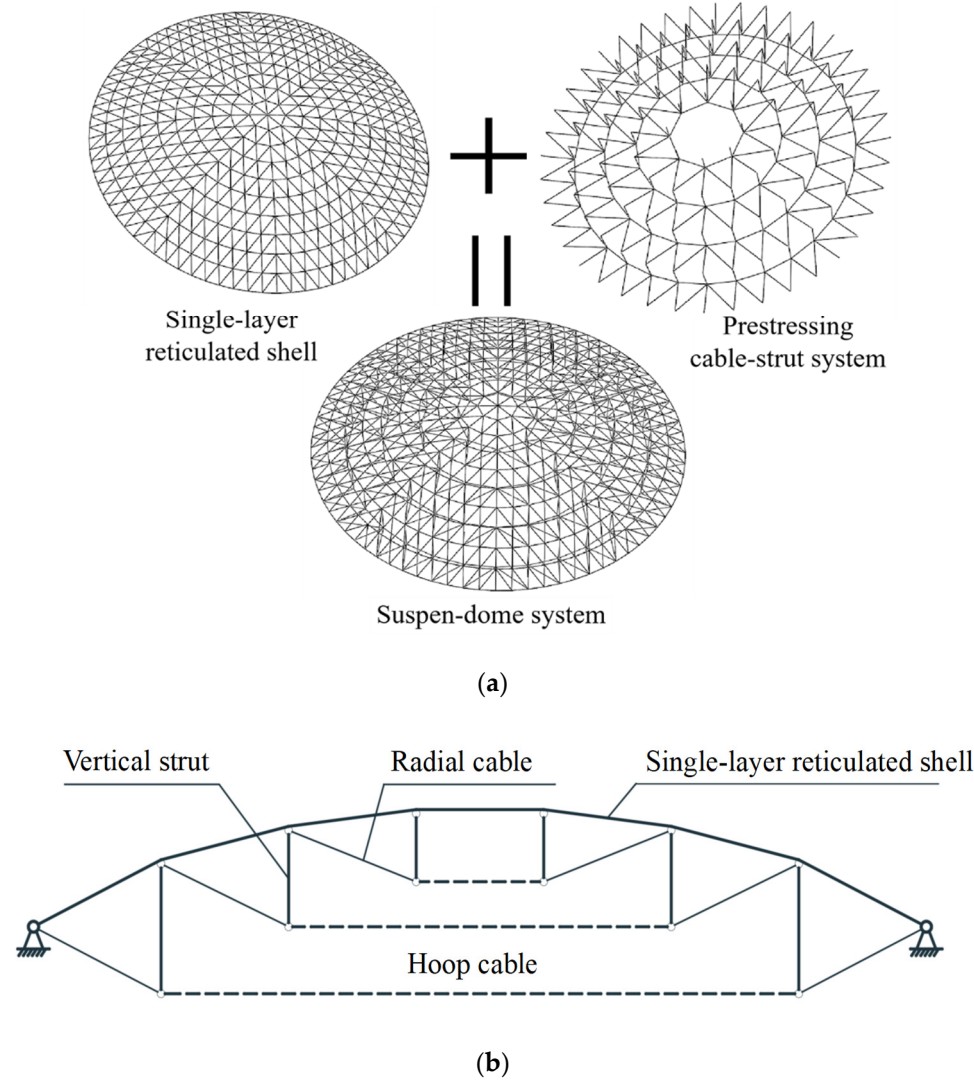

(**a**)

(**b**)

**Figure 1.** Suspen-dome structure: (**a**) Basic composition of suspen-dome structure; (**b**) Sectional view of the structure.

In the past few decades, with the wide application of suspen-dome structures in various major engineering projects, including stadiums, gymnasiums, and exhibition centers [4,5], scholars have conducted many studies on suspen-dome structures. In terms of structural static performance, Guo et al. [6] studied the worst distribution and magnitude of initial geometrical imperfection during stability calculation of the suspen-dome structure by the consistent imperfection mode method. Guo et al. [7] investigated the static performance of the suspen-dome structure under the heap load by a static test; the result proved that the heap load is more important to the structure than the full-span load. In terms of structural optimal design, Kaveh et al. [8,9] developed an algorithm for optimum design of suspen-dome structures considering the topology, geometry, and size of member sections using the cascade-enhanced colliding bodies optimization method. Zhang et al. [10] utilized ANSYS software to compile a set of prestress design procedures for suspen-dome structures. The self-internal force mode and the prestress level ratio among three-ring cables were analyzed by these procedures, and the prestress in the cable was determined. For constructional

technologies, Kitipornchai et al. [11] investigated the effects of initial geometric imperfections, rise-span ratio, and joint stiffness on the stable buckling capacity of the suspen-dome structure. A linear superposition analysis method that can accurately calculate the contribution of cable pretension and external load to component stress under different load cases was also proposed. Li et al. [12] developed a health-monitoring system that can achieve multitudinous physical variable synchronization acquisition to guarantee safety during the construction process of the suspen-dome structure. In terms of structural innovation, Xue et al. [13,14] proposed a new system of loop-free suspen-dome structures; for this system, the influence of cable removal on the static performance of the structure was deeply studied, and the good performance of the loop-free suspen-dome structure was proved.

In addition, since suspen-dome structures are often used as large public buildings, their dynamic characteristics and seismic response are essential to structural safety during earthquakes. Therefore, model tests are often needed to explore the seismic performance of suspen-dome structures. The most commonly used structural seismic performance test methods mainly include the seismic pseudo-static test, pseudo-dynamic test, and earthquake simulation shaking table test [15–18]. Among them, the pseudo-static test is essentially static loading, so it cannot reflect real earthquake action. The pseudo-dynamic test must have the means of timely calculation and data processing, an accurate test control method, and a high-precision automatic measurement system, and these conditions can only be realized by a computer and an electro-hydraulic servo test system device. Meanwhile, the pseudo-dynamic test is performed by the static test method, which will be different from the actual earthquake action. By contrast, the earthquake simulation shaking table test can realistically reproduce the earthquake process and is the most direct method to study the seismic response and failure mechanism of structures in the laboratory. Although the shaking table test is usually limited to small-scale model tests due to the limitations of its table bearing capacity and size, it is still the ideal test method to reproduce the ground motion and structural response [19]. Therefore, it has also become one of the important means to explore the seismic performance of suspen-dome structures. Wang et al. [20] produced a 1:10 scale model based on the 2008 Olympic Badminton Hall suspen-dome structure, and a shaking table test to study the dynamic characteristics and seismic responses of the structure was conducted; the results show that the suspen-dome structure has dense vibration modes and superior seismic performance. Lin et al. [21] investigated the dynamic characteristics and seismic response of the double elliptical suspen-dome structure by using a shaking table test, and the test results obtained are in good agreement with the theoretical results. Wu et al. [22] took a suspen-dome structure as the prototype, carried out a multi-dimensional earthquake simulation test on the scale model using eight shaking tables, and the structure's seismic performance under a multi-dimensional earthquake was researched with the improved plant growth simulation algorithm.

It is not difficult to find that in previous seismic performance tests, the research objects were all suspen-dome structures without a center-hung scoreboard. However, with the continuous increase of the audience's demand for game viewing, the center-hung scoreboard has become an indispensable large display device in modern indoor dome gymnasiums [23].

*1.2. Introduction*

Indoor dome gymnasiums are commonly used venues for various sports events, generally consisting of an upper roof structure, a lower spectator stand, and a supporting structure. The center-hung scoreboard, which can synchronously broadcast the game and capture its beautiful moments, is usually flexibly suspended from the roof of the indoor dome gymnasium. The center-hung scoreboard of the Prudential Center in New Jersey, USA is currently the largest in-arena scoreboard in the world [24]. The Wukesong Basketball Gymnasium, constructed for the 2008 Olympic Games, which was also the Ice Hockey Hall venue for the 2022 Winter Olympics, is the first stadium in China to install a full-color, high-definition scoreboard [25].

On the other hand, due to their excellent mechanical properties, more and more suspendome structures have been applied to the roof structures of long-span gymnasiums, especially in East Asia [26–31]. Among them, the Pingshan Basketball Gymnasium for the 2011 World University Games, spanning 72 m, is China's first suspen-dome structure to install a center-hung scoreboard [32], as shown in Figure 2a. The Dalian Gymnasium, with a span of 145.5 m × 116 m, is the largest suspen-dome structure with a center-hung scoreboard in Asia [33]. The center-hung scoreboard weighs more than 20 t, as shown in Figure 2b.

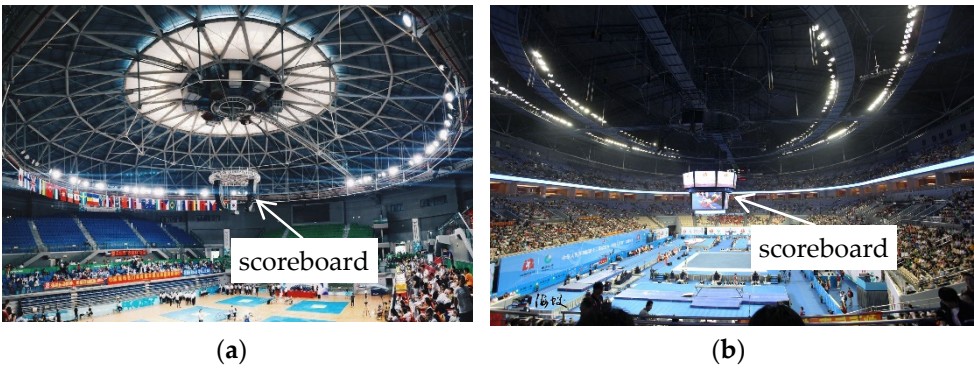

**Figure 2.** Suspen-dome structure with center-hung scoreboard: (**a**) Pingshan Basketball Gymnasium; (**b**) Dalian Gymnasium.

However, up to now, there have been no regulations and specifications for hanging scoreboards on suspen-dome roof structures. Center-hung scoreboards generally have considerable weight, some of which weigh more than 20 t [34]. Therefore, when the scoreboard is suspended from the suspen-dome structure, a large, concentrated suspension load will be formed in the center of the upper reticulated shell. Meanwhile, the additional inertia effect of the scoreboard on the structure may also have a non-negligible impact on the dynamic characteristics and seismic response of the suspen-dome structure. At present, no literature has been found related to the research on the seismic performance of suspen-dome structures with center-hung scoreboards. Thus, it is necessary to research the influence of the center-hung scoreboard on the dynamic characteristics and seismic response of the suspen-dome structure.

To this end, this paper took the suspen-dome structure with a center-hung scoreboard of the Lanzhou Olympic Gymnasium as the research object, and a dynamic scale model with a geometric similarity ratio of 1:20 was established. Then, the white noise excitation test and numerical modal analysis were carried out for the models with and without the center-hung scoreboard, respectively; the natural frequencies and mode shapes of the two models were tested, and the influence of the center-hung scoreboard on the natural vibration characteristics of the suspen-dome structure was investigated. Furthermore, various seismic responses, such as acceleration response, displacement response, and strain response, of the two models with and without a center-hung scoreboard were tested by conducting an earthquake simulation test. The effect of the center-hung scoreboard on the seismic response of the suspen-dome structure was studied. Finally, the acceleration response and displacement response of the center-hung scoreboard were tested and analyzed. This paper will provide a reference for further study on the dynamic characteristics and seismic performance of suspen-dome structures with center-hung scoreboards. Moreover, the research results of this paper have practical significance for the application of center-hung scoreboards in suspen-dome structures and the prevention of the adverse effects of scoreboards on the seismic performance of the structure.

## 2. Shaking Table Test Design

### 2.1. Engineering Background

The Lanzhou Olympic Sports Center is located in Lanzhou, Gansu Province, China. The gymnasium roof adopts a suspen-dome structure with a span of 94 m and a rise of 8 m, as shown in Figure 3. The suspen-dome structure is mainly composed of the upper ribbed-type single-layer reticulated shell and the lower Levy-type cable-strut system, as shown in Figure 4a. The upper single-layer reticulated shell contains 528 members and 253 nodes; the radial and ring bars are the main components of the upper reticulated shell structure, the cross-section type of which is the box-shaped section; the slant bars are used to increase the in-plane stiffness of the reticulated shell, the cross-section type of which is a circular pipe; these bars and nodes are all made of Q355B steel. The lower Levy-type cable-strut system consists of five layers, including five hoop cables, 186 radial cables, and 90 vertical struts. The upper ends of the struts of each ring are hinged with the nodes corresponding to the single-layer reticulated shell. The lower ends of the struts are connected with the nodes of the next ring of the single-layer reticulated shell by radial cables, and the lower ends of the struts of the same ring are connected by the hoop cables, so that the whole structure forms a complete system, as shown in Figure 4b,c. The Galfan-coated steel strand has the advantages of strong prestress, anti-corrosion, anti-friction and fire protection, so it is used for the hoop cables; the Q690D steel not only has very high strength but also has certain flexibility, wear resistance, fatigue resistance, and impact resistance, so the radial cables adopt solid circular bars with Q690D steel; the Q355B steel pipe has high strength, high precision, long pipe section and few sockets, so it is used for the reticulated shell members and vertical struts. The detailed information for all components is listed in Table 1. The seismic fortification intensity of the project is 8 degrees, the design basic seismic acceleration is 0.2 g, the construction site classification is class II, and the design earthquake group is the third group. The suspen-dome structure of the Lanzhou Olympic Gymnasium belongs to a typical long-span spatial structure, and a center-hung scoreboard is suspended from the roof by four slings, as shown in Figure 4c. The center-hung scoreboard consists mainly of several LED display screens and the steel construction to arrange these LED display screens, with a total weight of up to 30 t.

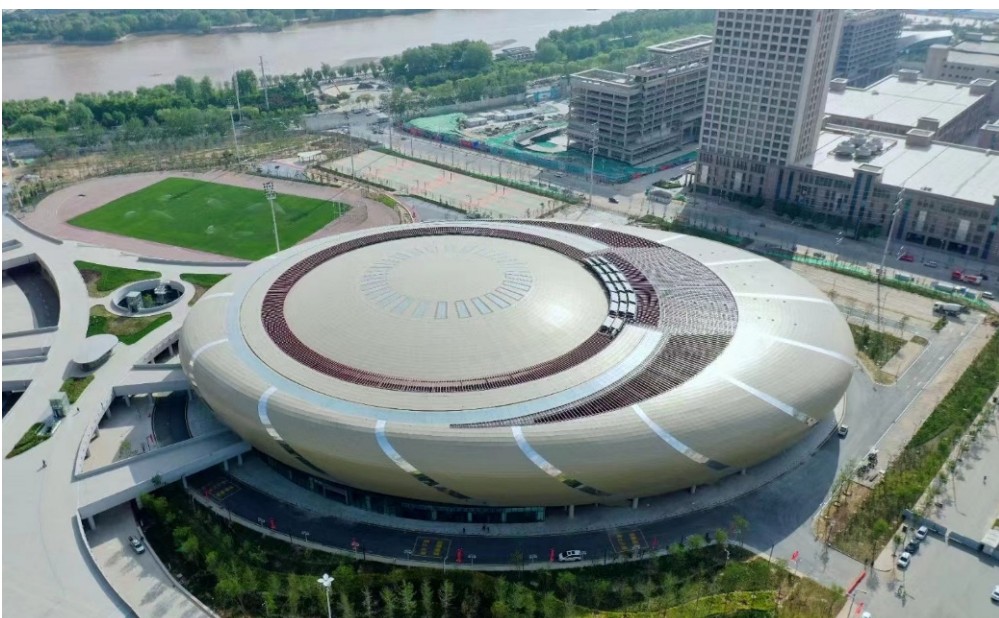

**Figure 3.** The Gymnasium of Lanzhou Olympic Sports Center.

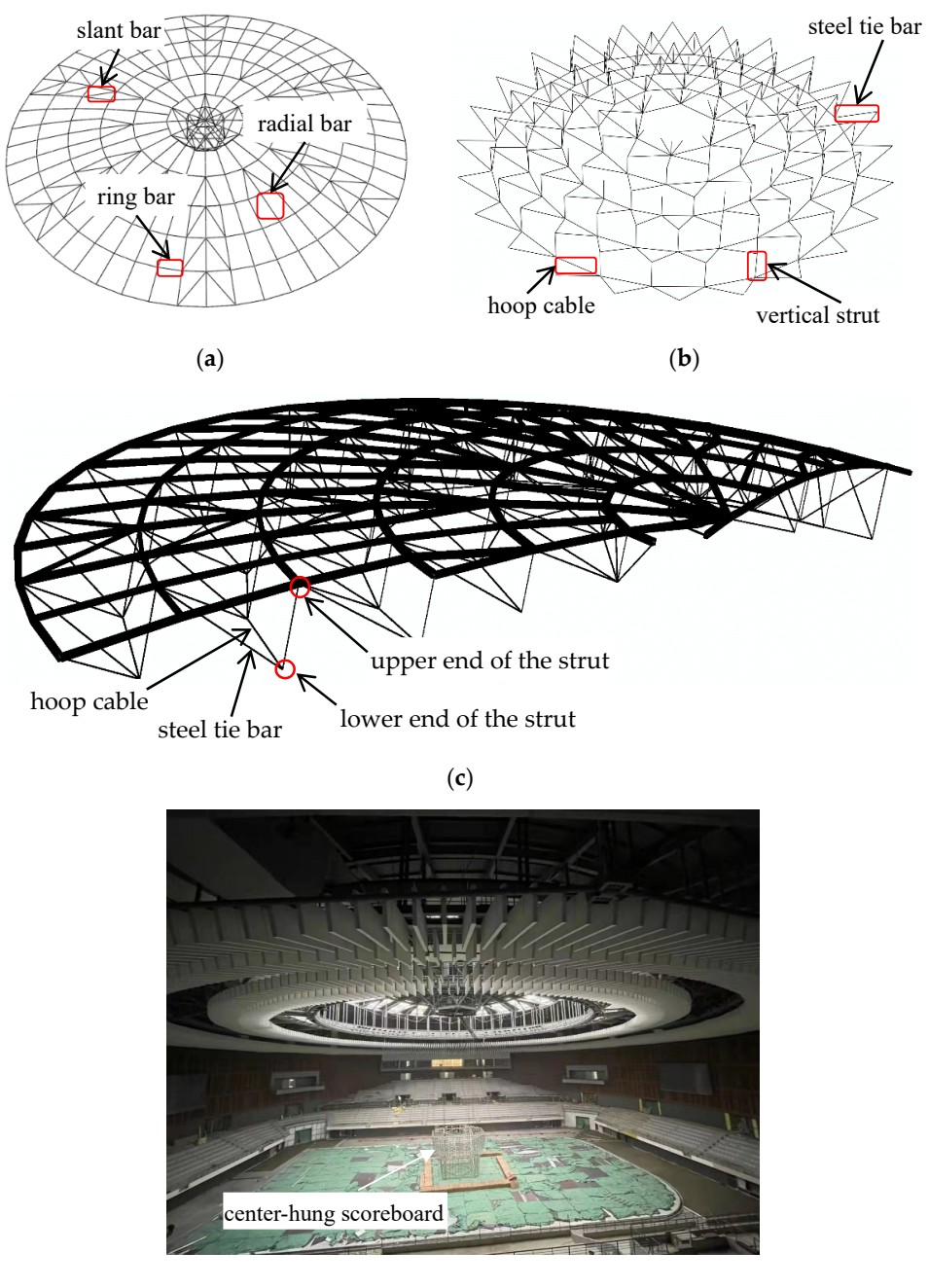

**Figure 4.** The suspen-dome structure of Lanzhou Olympic Sports Center: (**a**) Single-layer reticulated shell; (**b**) Cable-strut system; (**c**) The composition of the structure; (**d**) The center-hung scoreboard.

**Table 1.** Information of components.

| Member Type | | Quantity | Section Shape | Material |
|---|---|---|---|---|
| Reticulated shell | Radial bar | 276 | Box-shaped | Q355B steel |
| | Ring bar | 252 | Box-shaped | Q355B steel |
| | Node | 253 | – | Q355B steel |
| Cable-strut system | Radial cable | 186 | Solid circular | Q690D steel |
| | Hoop cable | 5 | Strand | Galfan-coated steel |
| | Vertical strut | 90 | Circular pipe | Q355B |
| | Cable-strut node | 90 | – | Q355B |

### 2.2. Test Model Design and Fabrication

Since the size of the shaking table in the test is 5.0 m × 5.0 m, the geometric similarity ratio between the scale model and the prototype structure is finally determined to be 1:20, and the diameter of the scale model is 4.7 m. However, when designed according to the geometric similarity ratio, the sections of the model members are small and thin and difficult to obtain. For this reason, Austenitic 304 stainless steel was selected to replace Q355B steel to solve the above problems. The fundamental mechanical performance parameters of Austenitic 304 stainless steel and Q355 steel were measured through the material property test, as shown in Table 2. The stress similarity ratio of the material was set to 1:1.5 according to the ratio of the yield strength $f_y$ of the two materials. The members and nodes of the actual test model are shown in Figure 5a–c. Among them, radial bars and ring bars adopt box sections, and slant bars adopt tube sections. The preload of the hoop cables from the inside to the outside were 0.56 kN, 0.55 kN, 1.25 kN, 2.19 kN, and 4.73 kN, respectively, which is applied by tightening the sleeve of the steel tie bar, as shown in Figure 5d. The overall model is shown in Figure 5e.

**Table 2.** Basic mechanical properties of two types of steel.

| Steel Type | $E$/MPa | $f_y$/MPa | $f_u$/MPa |
|---|---|---|---|
| Austenitic 304 | 204,000 | 238 | 870 |
| Q355B | 206,000 | 355 | 470 |
| Ratio (304/Q355) | 1.01 | 1.5 | / |

The bars of the center-hung scoreboard are made of the same material as the reticulated shell members, which are connected to the hanging platform in the center of the reticulated shell by four slings. In order to simulate the actual weight of the scoreboard in the prototype structure, mass blocks were placed inside the scoreboard in the test model, as shown in Figure 6, and the total weight of the center-hung scoreboard and mass blocks of the scale model is 0.5 kN.

The lower part of the model is a steel column-ring beam supporting system, all made of Q345 steel; the ring beam is a box-shaped section with a side length of 50 mm and a thickness of 2 mm, the steel columns and diagonal braces are circular pipe sections with a diameter of 48 mm and a thickness of 2.8 mm, and the height of the steel column is 0.98 m. The bottom of the column was welded to a steel base similar to a rigid body, and M20 bolts were used to connect the base to the shaking table. Since Asia, especially China, is located at the intersection of the two major seismic zones in the world, the Pacific Rim seismic zone and the Eurasian seismic zone, earthquake disasters are prone to occur [35]. Meanwhile, according to the Chinese Code [36,37], the dead and live roof loads of the prototype structure were designed as 1.2 kN/m$^2$ and 0.5 kN/m$^2$, respectively. After considering the similarity ratio of 1:1.5, the test model roof's dead and live loads were 0.8 kN/m$^2$ and 0.33 kN/m$^2$, respectively. All loads were set by applying mass blocks on the model nodes or in the middle of bars, as shown in Figure 7a,b, and the assembled model is shown in Figure 7c.

Combined with the scale model's geometric similarity ratio of 1:20 and the stress similarity ratio of 1:1.5, the acceleration similarity ratio was set to 1:1. The above three similarity ratios were taken as controllable similarity constants, and the similarity constants of other physical quantities can be determined by the dimensional analysis method [38]. The similarity constants of the scale model's main physical parameters are listed in Table 3.

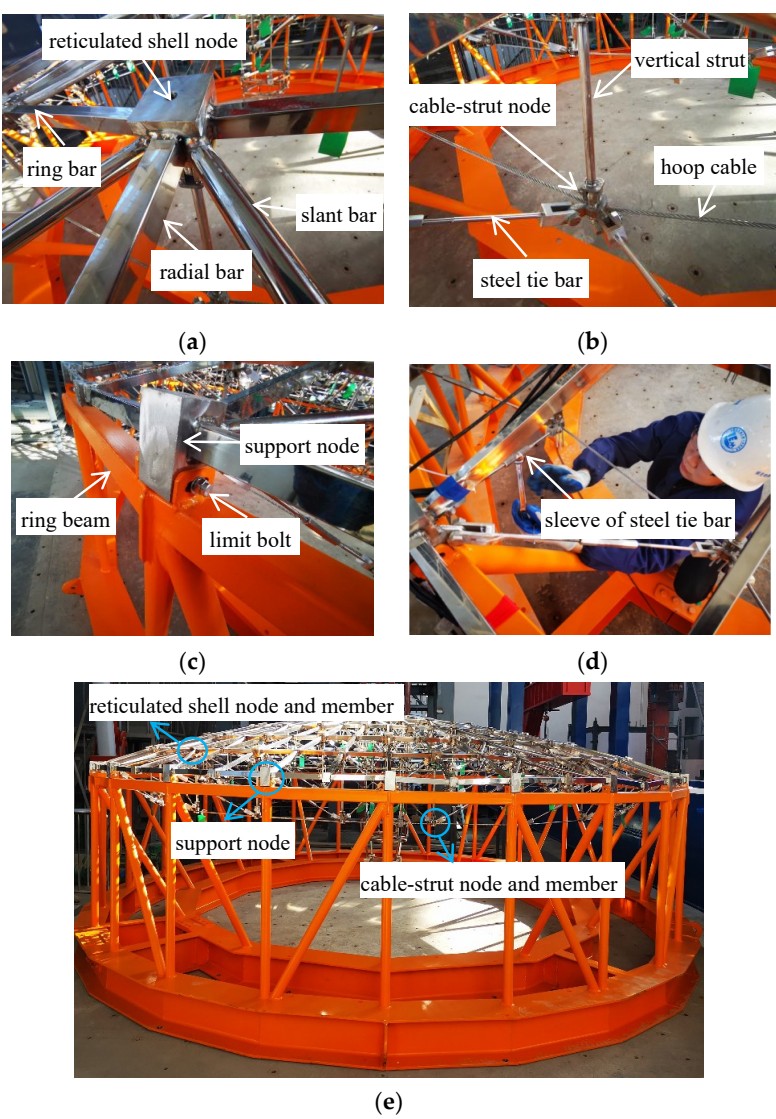

**Figure 5.** Nodes and Members: (**a**) Reticulated shell node and member; (**b**) Cable-strut node and member; (**c**) Support node; (**d**) Applying preload; (**e**) Overall model.

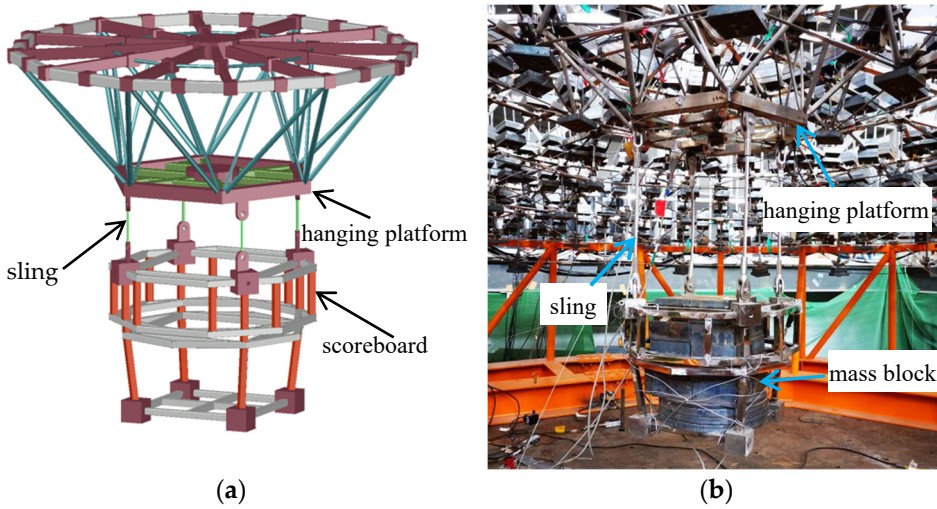

**Figure 6.** Center-hung scoreboard: (**a**) Scoreboard structural drawing; (**b**) Scoreboard test model.

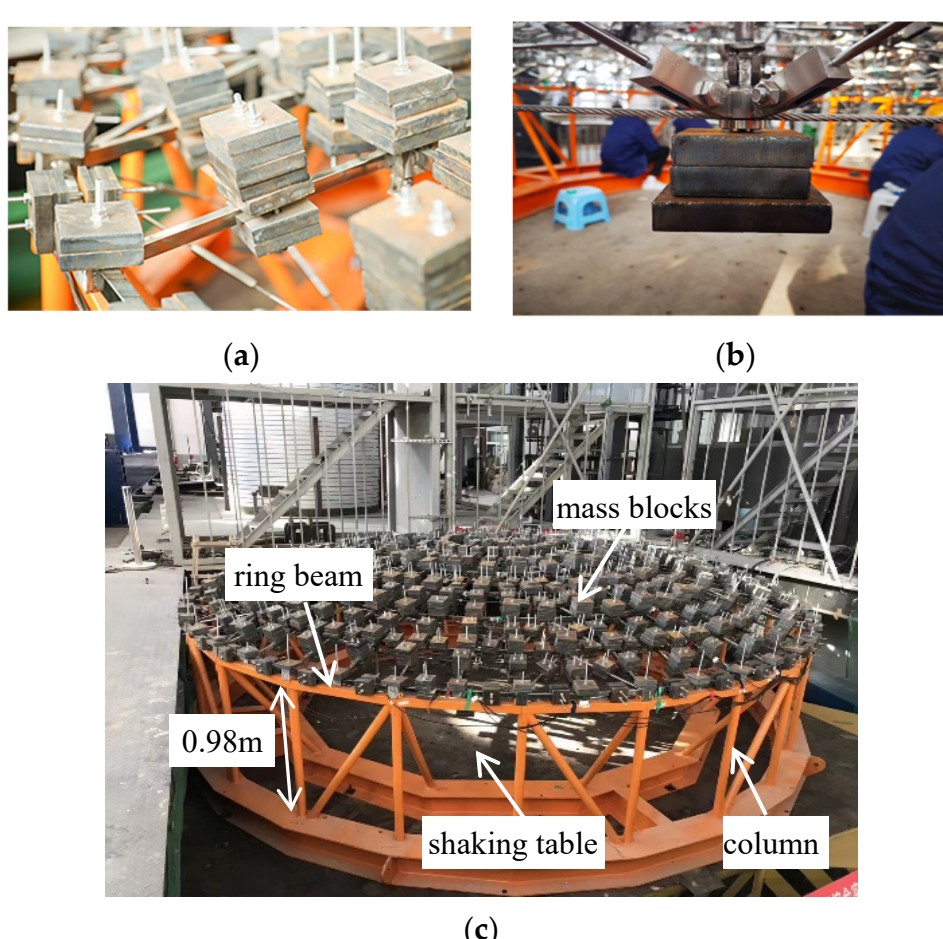

**Figure 7.** Test scale model: (**a**) Mass blocks of reticulated shell; (**b**) Mass blocks of cable-strut node; (**c**) Assembled model.

**Table 3.** The similarity constants of the main physical parameters of the scale model.

| Physical Properties | Physical Parameter | Symbol | Scaling Factors |
|---|---|---|---|
| | Length | $S_L$ | 1:20 |
| Geometric | Area | $S_A$ | 1:400 |
| | Displacement | $S_Z$ | 1:30 |
| | Strain | $S_\varepsilon$ | 1:1 |
| | Elastic Modulus | $S_E$ | 1:1 |
| Material | Stress | $S_\sigma$ | 1:1.5 |
| | Poisson's ratio | $S_\mu$ | 1:1 |
| | Mass density | $S_\rho$ | 20:1.5 |
| | Concentration | $S_P$ | 1:600 |
| Load | Surface load | $S_q$ | 1:1.5 |
| | Period | $S_T$ | $1:\sqrt{20}$ |
| Power | Frequency | $S_f$ | $\sqrt{20}:1$ |
| | Acceleration | $S_a$ | 1:1 |
| | Acceleration of gravity | $S_g$ | 1:1 |

## 2.3. Layout of Measuring Points

According to the model's symmetry and the test site's conditions, measuring points are arranged on the reticulated shell, hoop cables, and center-hung scoreboard to install the corresponding acquisition instruments. Figure 8a–g depicts the specific measuring point layout. The measuring points are given in the form of "xa-b". "x" is the code of the test content, including A, D, C, R, and HS, representing acceleration, displacement, ring bar

strain, radial bar strain, and hoop cable strain; "a" is the radial number of the measuring point, which is numbered from 1 to 48, as shown in Figure 8a; "b" is the ring number of the measuring point, which is numbered from 1 to 7 from the inner ring to the outer ring of the reticulated shell, as shown in Figure 8a. The measuring point of the reticulated shell center is "x1-0". The measuring points of the center-hung scoreboard are shown in Figure 8g.

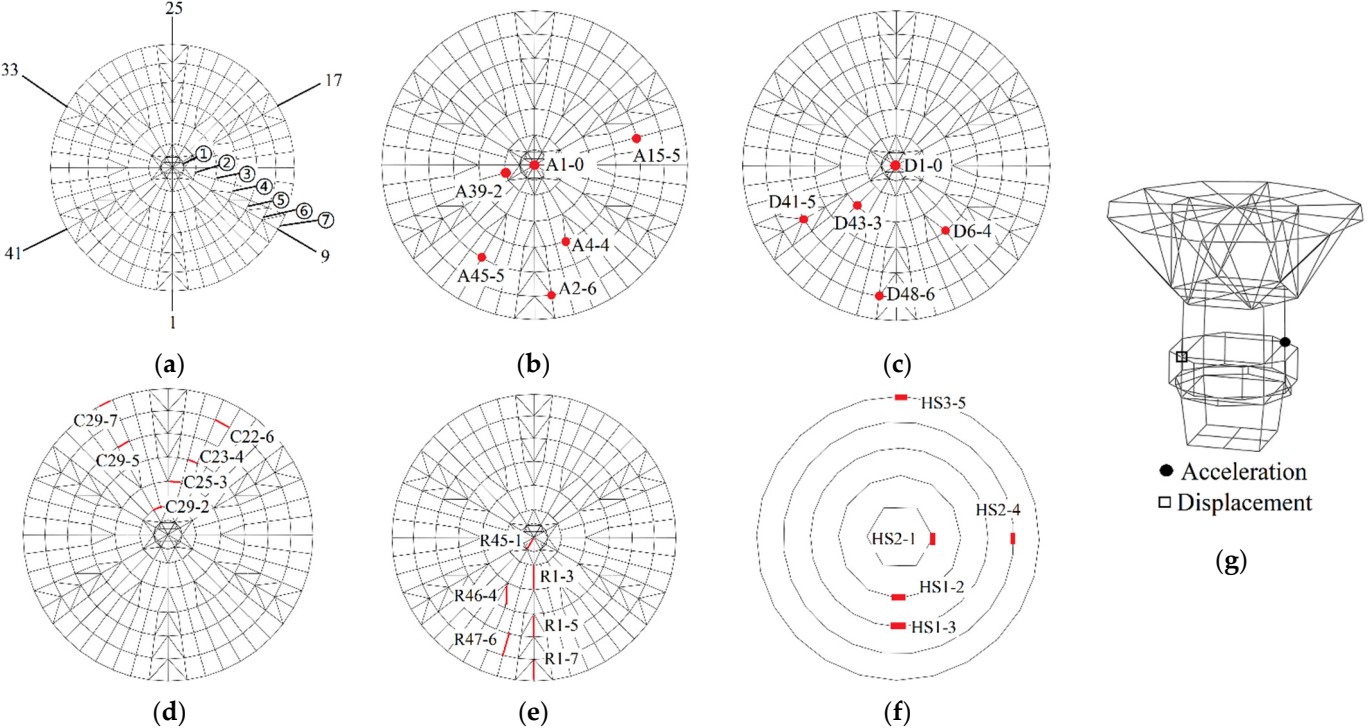

**Figure 8.** Measuring point layout: (**a**) Radial number and ring number; (**b**) Measuring point of acceleration; (**c**) Measuring point of displacement; (**d**) Measuring point of ring bar; (**e**) Measuring point of radial bar; (**f**) Measuring point of hoop cable; (**g**) Measuring point of center-hung scoreboard.

In the test, an acceleration sensor with a range of $-160\sim160$ m/s$^2$ was used to collect the acceleration response data of the model, a pull-wire displacement sensor with a range of $250\sim1000$ mm was used to collect displacement response data, and the member strain data was collected by strain gauges with a resistance value of $120.3 \pm 0.1\ \Omega$ and a sensitivity coefficient of $2.22 \pm 1\%$. All acquisition instruments are presented in Figure 9.

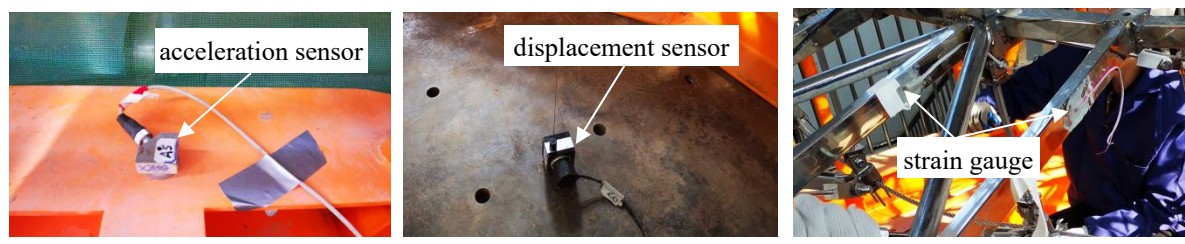

**Figure 9.** Acquisition instruments.

## 3. Analysis of Natural Vibration Characteristics

### 3.1. Natural Frequency

An equal-intensity white noise excitation test was conducted on the models with and without the center-hung scoreboard. Subsequently, the natural frequencies of the two modes were identified by performing spectral analysis of the acquired acceleration data. The comparison of the first six-order natural frequencies between the two modes is listed in Table 4. It can be seen that the fundamental frequency of the model with the center-hung

scoreboard is much lower than that of the model without the center-hung scoreboard. According to Table 4, the 3rd, 4th, and 6th order natural frequencies of the model with the scoreboard are almost equal to the first 3-order frequencies of the model without the scoreboard, while the 1st, 2nd, and 5th order natural frequencies of the model with the scoreboard are quite different from the corresponding order's natural frequencies of the model without the scoreboard. Meanwhile, during the test, in addition to the vibration of the reticulated shell, only the swing of the scoreboard was remarkable. Therefore, it can be inferred that under the excitation of white noise, the swing of the center-hung scoreboard is first excited as the first-order mode shape; and then the 1st, 2nd, and 5th order vibration modes of the model with the scoreboard are all due to the swing of the scoreboard. The above inference will also be further confirmed in the following section.

**Table 4.** First six-order natural frequencies.

| Order | Without Scoreboard/Hz | With Scoreboard/Hz |
|---|---|---|
| 1 | 5.871 | 1.367 |
| 2 | 5.871 | 1.566 |
| 3 | 9.393 | 5.871 |
| 4 | 11.150 | 5.871 |
| 5 | 11.350 | 7.436 |
| 6 | 11.941 | 9.198 |
| Fundamental frequency | 5.871 | 1.370 |

In order to further investigate the influence of the center-hung scoreboard on the natural frequency of the overall suspen-dome structure, the first 15-order natural frequencies of the two models were compared in Figure 10, excluding the center-hung scoreboard's vibration modes. It can be found that the difference in the lower-order natural frequencies between the two is not apparent. By contrast, the natural frequencies of the 8th to 15th orders of the model with the center-hung scoreboard are evidently higher than those of the model without the center-hung scoreboard. That is to say, the center-hung scoreboard can increase the higher-order natural frequencies of the overall structure.

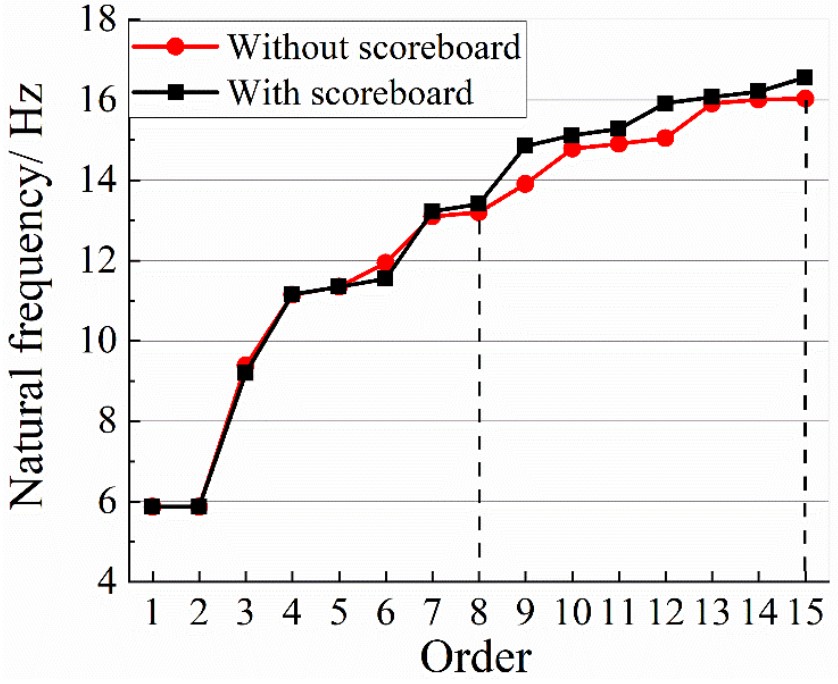

**Figure 10.** First 15-order natural frequencies of the two overall models.

### 3.2. Model Shape

#### 3.2.1. Finite Element Model

The numerical models with and without the center-hung scoreboard were established by ABAQUS software, as shown in Figure 11. Table 5 lists the specific parameters of the numerical model members. The B31 three-dimensional beam elements were selected to simulate the reticulated shell and substructure members, and T3D2 truss elements were selected to simulate the members of the cable-strut system. Each reticulated shell member was divided into two meshes; each cable-strut member was set to one mesh. The connection between the upper structure and the ring beam released the radial constraint. The members of the reticulated shell utilized fixed connections, the members of the cable-strut system were hinged, and the upper end of the strut and the steel tie bar were all hinged with the reticulated shell. The roof load was supposed to be mass agglomerated on the nodes of the model. In the test, the coordinates of each node were measured, and the measurement results were taken as the final coordinates of the finite element model's nodes after considering geometric defects. For the center-hung scoreboard, the B31 elements simulated the steel construction, and the T3D2 elements simulated the four slings. Each of the scoreboard members and each sling were set to one mesh. The additional mass of the center-hung scoreboard was achieved by applying a concentrated mass element at the midpoint of its bottom. Since only modal analysis was performed on the two numerical models, the yield strength of the material was not considered.

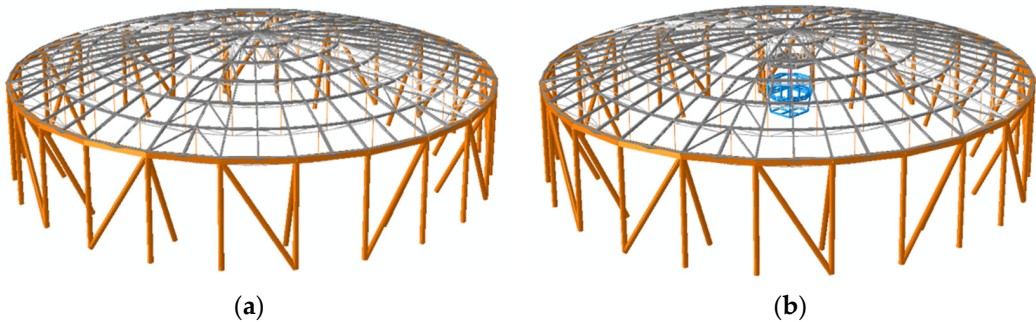

|      |      |
| :--: | :--: |
| (**a**) | (**b**) |

**Figure 11.** Numerical model: (**a**) Without center-hung scoreboard; (**b**) With center-hung scoreboard.

**Table 5.** Parameters of numerical model members.

| Member Type | | Element Type | Mesh Number | Material Elastic Modulus/Mpa |
| :---: | :---: | :---: | :---: | :---: |
| Reticulated shell | Radial bar | B31 | 2 | 204,000 |
| | Ring bar | B31 | 2 | 204,000 |
| | Steel tie bar | T3D2 | 1 | 206,000 |
| Cable-strut system | Hoop cable | T3D2 | 1 | 160,000 |
| | Vertical strut | T3D2 | 1 | 206,000 |
| Center-hung scoreboard | Sling | T3D2 | 1 | 160,000 |
| | Main member | B31 | 1 | 204,000 |

#### 3.2.2. Modal Analysis

The block Lanczos method was used to conduct modal analysis of the two numerical models with and without the center-hung scoreboard, and the first six-order mode shapes of the two models were compared, as shown in Table 6. It can be observed that the first-order, the second-order, and the fifth-order vibration modes of the structure with the center-hung scoreboard are the mode shapes of the center-hung scoreboard, which are the swing in the x-direction, the swing in the y-direction, and the rotation around the z-axis. Meanwhile, the order of the mode shapes of the center-hung scoreboard and the natural frequencies of the numerical model are basically consistent with the test results, which illustrates the accuracy of the numerical model. In addition, the overall mode shapes of the two models

are both dominated by vertical vibration, indicating that the center-hung scoreboard has little effect on the lower-order overall mode shape of the suspen-dome structure.

**Table 6.** Comparison of mode shapes.

| Mode Shape Type | | Without Scoreboard | With Scoreboard |
|---|---|---|---|
| The first-order mode shape of center-hung scoreboard | Mode shape (Frequency) | \ |  (1.429 Hz) |
| | Order | \ | 1 |
| The second-order mode shape of center-hung scoreboard | Mode shape (Frequency) | \ |  (1.507 Hz) |
| | Order | \ | 2 |
| The third-order mode shape of center-hung scoreboard | Mode shape (Frequency) | \ |  (7.084 Hz) |
| | Order | \ | 5 |
| The first-order mode shape of overall model | Mode shape (Frequency) |  (5.749 Hz) |  (5.764 Hz) |
| | Order | 1 | 3 |
| The second-order mode shape of overall model | Mode shape (Frequency) |  (5.796 Hz) |  (5.818 Hz) |
| | Order | 2 | 4 |
| The third-order mode shape of overall model | Mode shape (Frequency) |  (9.419 Hz) |  (9.300 Hz) |
| | Order | 3 | 6 |

From the above results, it can be deduced that the center-hung scoreboard will affect the natural vibration characteristics of the suspen-dome structure. The center-hung scoreboard's swing will be excited first as the low-order mode shape. Moreover, the higher-order natural frequencies of the suspen-dome structure will be increased when the scoreboard is suspended.

## 4. Analysis of Earthquake Simulation Test Results

### 4.1. Loading Scheme

In order to investigate the impact of the center-hung scoreboard on the seismic performance of the suspen-dome structure, the various seismic responses of the models with and without a center-hung scoreboard under seismic loads were compared by conducting an earthquake simulation test. Three seismic waves, including the Taft wave, Hollister wave, and Cholame wave, were selected for the test, and then the length of time for them was 21.0 s, 22.9 s, and 14.8 s, respectively. The amplitude of all the seismic waves is modified to 0.2 g and input along the x-direction at the bottom of the column. The acceleration time-history curves of the three seismic waves are shown in Figure 12a–c. Table 7 is the loading scheme, and the white noise excitation test was carried out after each case was completed.

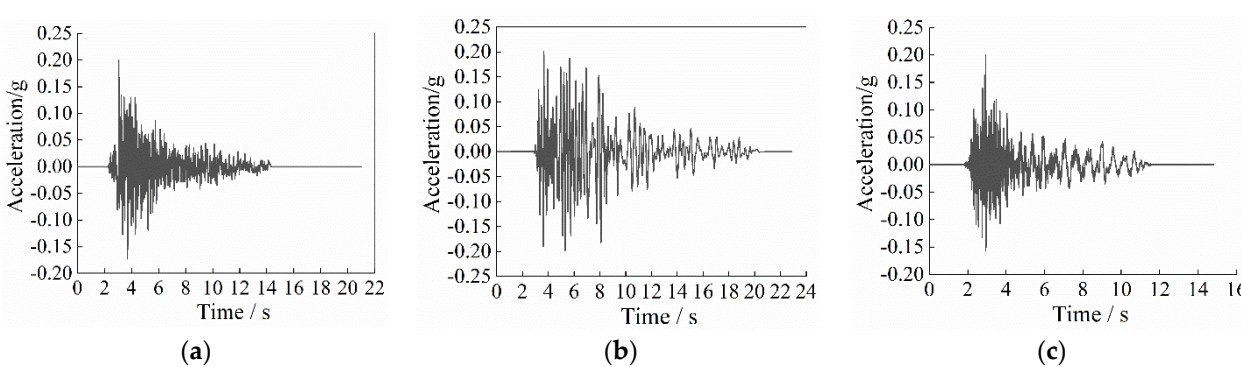

**Figure 12.** Seismic wave records: (**a**) TAFT wave acceleration time-history curve; (**b**) Hollister wave acceleration time-history curve; (**c**) Cholame wave acceleration time-history curve.

**Table 7.** Loading case for shaking table test.

| Case | Seismic Wave | Amplitude | Model |
|---|---|---|---|
| 1 | Taft | 0.2 g | without scoreboard |
| 2 | White noise | 0.1 g | |
| 3 | Taft | 0.2 g | with scoreboard |
| 4 | White noise | 0.1 g | |
| 5 | Hollister | 0.2 g | without scoreboard |
| 6 | White noise | 0.1 g | |
| 7 | Hollister | 0.2 g | with scoreboard |
| 8 | White noise | 0.1 g | |
| 9 | Cholame | 0.2 g | without scoreboard |
| 10 | White noise | 0.1 g | |
| 11 | Cholame | 0.2 g | with scoreboard |
| 12 | White noise | 0.1 g | |

### 4.2. Analysis of Acceleration Response

#### 4.2.1. Acceleration Response Comparison of the Models with and without Center-Hung Scoreboard

Figure 13a shows the peak acceleration responses of the two models; Figure 13b shows the acceleration time-history curves of the measuring point A1-0 in the z-direction. It can be seen that the acceleration responses of almost all the measuring points on the model with the center-hung scoreboard are higher than those of the model without the center-hung scoreboard. Under the action of the TAFT wave, the peak accelerations in the x-, y-, and z-directions of the model with the scoreboard increased by 0.06~0.4 g, 0.01~0.17 g, and 0.02~1.01 g, respectively, compared to the model without the scoreboard; under the action of the Hollister wave, the peak accelerations in the three directions of the model with the scoreboard increased by 0.03~0.38 g, 0.04~0.18 g, and 0.14~0.56 g, respectively; under the action of the Cholame wave, the peak accelerations in the three directions of the model

with the scoreboard increased by 0.05~0.34 g, 0.01~0.16 g, and 0.02~0.49 g, respectively, compared to the model without the scoreboard. It can also be seen from Figure 13a that the acceleration increase of measuring point A1-0 located in the reticulated shell center is the most pronounced. The above results indicate that the center-hung scoreboard can significantly increase the acceleration of the suspen-dome structure.

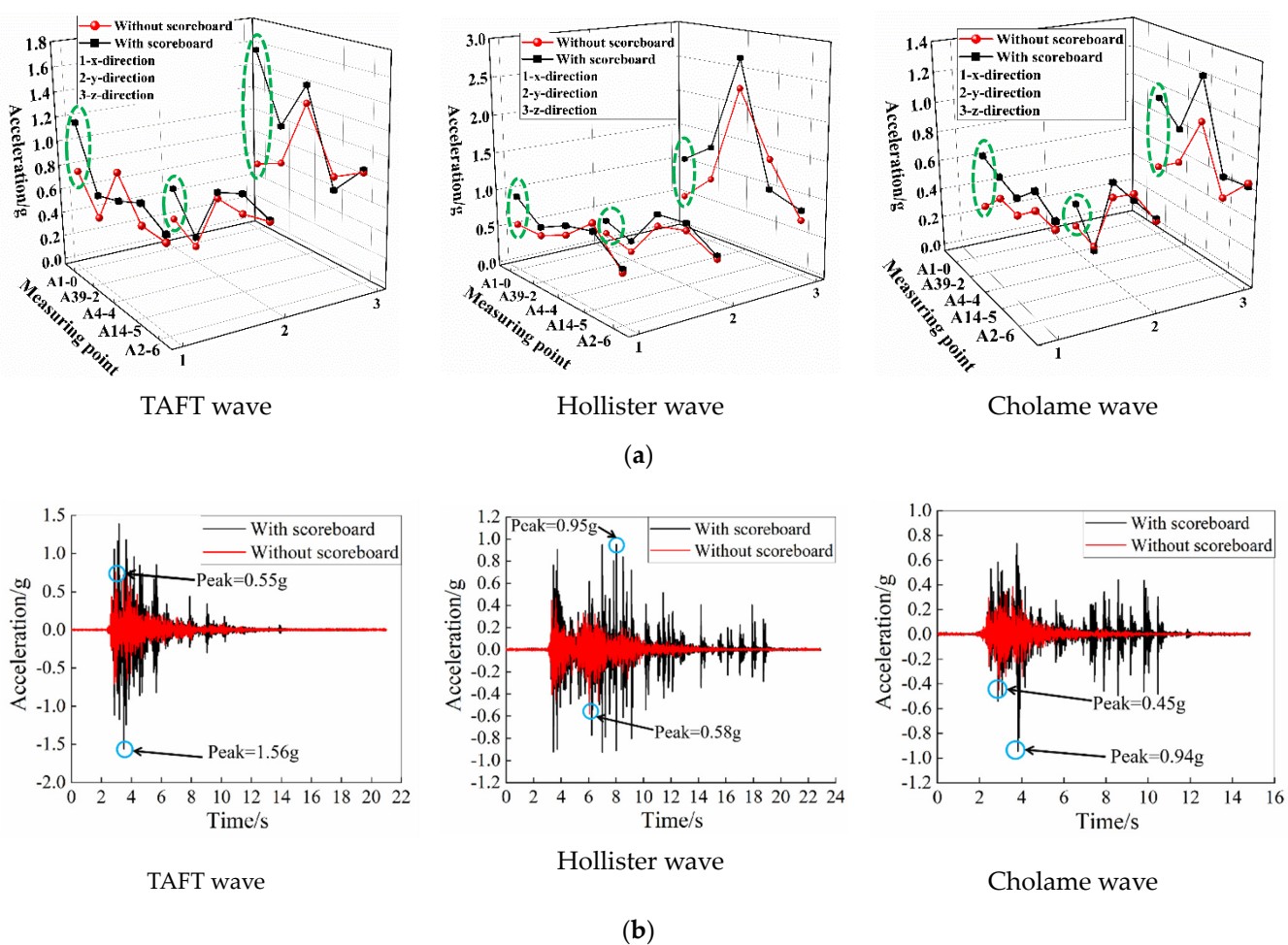

**Figure 13.** Acceleration response comparison: (**a**) Peak acceleration; (**b**) Acceleration time-history curves of measuring point A1-0 in the z-direction.

Table 8 compares the acceleration amplification coefficients along the seismic input direction (the x-direction) of measuring points A1-0, A4-4, and A2-6 of the two models. The ratio of the peak acceleration response $a$ of the structure to the input peak acceleration $a_g$ is defined as the acceleration amplification coefficient $R_a$:

$$R_a = a / a_g$$

It can be seen from the acceleration amplification coefficient that the model with the center-hung scoreboard has a more significant amplifying effect on the input seismic load than the model without the center-hung scoreboard in nearly all working conditions. Under the excitation of the Hollister wave, the acceleration amplification coefficients of the three measuring points of the model with the scoreboard can reach up to 4.32, 3.86, and 2.86, respectively, while those of the model without the scoreboard are only 2.71, 3.48, and 2.81, respectively, which further indicates that the center-hung scoreboard will increase the acceleration response of the suspen-dome structure.

**Table 8.** Comparison of acceleration amplification coefficient.

| Measuring Point | Seismic Wave | Model | Input Peak Acceleration ($a_g$)/g | Peak Acceleration Response ($a$)/g | $R_a = a/a_g$ |
|---|---|---|---|---|---|
| A1-0 | TAFT | without scoreboard | 0.23 | 0.78 | 3.39 |
| | | with scoreboard | 0.21 | 1.18 | 5.62 |
| | Hollister | without scoreboard | 0.21 | 0.57 | 2.71 |
| | | with scoreboard | 0.22 | 0.95 | 4.32 |
| | Cholame | without scoreboard | 0.19 | 0.41 | 2.16 |
| | | with scoreboard | 0.21 | 0.75 | 3.57 |
| A4-4 | TAFT | without scoreboard | 0.23 | 0.96 | 4.17 |
| | | with scoreboard | 0.21 | 0.74 | 3.52 |
| | Hollister | without scoreboard | 0.21 | 0.73 | 3.48 |
| | | with scoreboard | 0.22 | 0.85 | 3.86 |
| | Cholame | without scoreboard | 0.19 | 0.53 | 2.79 |
| | | with scoreboard | 0.21 | 0.64 | 3.05 |
| A2-6 | TAFT | without scoreboard | 0.23 | 0.65 | 2.82 |
| | | with scoreboard | 0.21 | 0.71 | 3.38 |
| | Hollister | without scoreboard | 0.21 | 0.59 | 2.81 |
| | | with scoreboard | 0.22 | 0.63 | 2.86 |
| | Cholame | without scoreboard | 0.19 | 0.63 | 3.32 |
| | | with scoreboard | 0.21 | 0.68 | 3.24 |

Figure 14 presents the relative error of the peak acceleration between the two models. It can be found that under the action of all three seismic waves, the relative error of the peak acceleration between the two shows a decreasing trend from the inner ring to the outer ring of the structure; taking the relative error of the acceleration in the z-direction between the two models as an example, when the model was excited by the TAFT wave, Hollister wave, and Cholame wave, respectively, the relative error of the peak acceleration of the central measuring point A1-0 between the two models reached 1.0 g, 0.55 g, and 0.5 g, those of the intermediate measuring point A4-4 are 0.15 g, 0.41 g, and 0.21 g, while those of the outermost ring measuring point A2-6 are merely 0.01 g, 0.15 g, and 0.01 g, respectively. That is to say, the center-hung scoreboard's influence on the structure's acceleration response is gradually reduced from the inner ring to the outer ring.

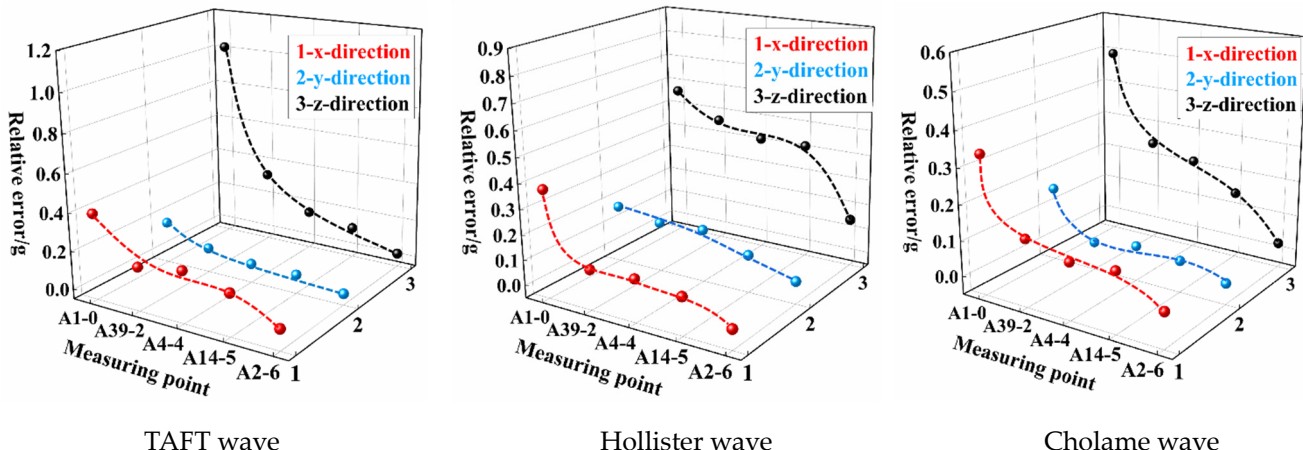

| TAFT wave | Hollister wave | Cholame wave |

**Figure 14.** Relative errors of peak acceleration between the two models.

### 4.2.2. Acceleration Response of Center-Hung Scoreboard

Table 9 shows the peak acceleration responses of the center-hung scoreboard. As the result shows, when a seismic wave is input, the acceleration of the scoreboard is extremely large and visibly higher than that of other measuring points. For instance, under the Hollister wave's action, the scoreboard's peak acceleration in three directions is up to 6.15 g, 4.42 g, and 7.41 g, respectively; while in Figure 12a, under the Hollister wave's action, the maximum peak accelerations in three directions of the reticulated shell are only 0.95 g,

0.77 g, and 2.65 g, respectively. The former is 6.47 times, 5.74 times, and 2.80 times the latter, respectively. That is to say, under the excitation of a seismic load, the center-hung scoreboard is prone to violent movement.

**Table 9.** Peak acceleration of center-hung scoreboard.

| Seismic Wave | Peak Acceleration/g | | |
|---|---|---|---|
| | X-Direction | Y-Direction | Z-Direction |
| Taft wave | 5.43 | 3.57 | 5.82 |
| Hollister wave | 6.15 | 4.42 | 7.41 |
| Cholame wave | 3.22 | 2.44 | 3.75 |

*4.3. Analysis of Displacement Response*

4.3.1. Displacement Response Comparison of the Models with and without Center-Hung Scoreboard

Figure 15a shows the peak displacement responses in the x-, y-, and z-directions of the two models; Figure 15b shows the displacement time-history curves of measuring point D1-0 in the z-direction. In order to make the comparison more apparent, all the displacements are relative values to the bottom of the column. It can be observed that the displacement responses of basically all of the measuring points of the model with the center-hung scoreboard are higher than those of the model without the center-hung scoreboard, and the difference between the two is undeniable in the x- and z-directions. Under the excitation of the TAFT wave, the peak displacements in the x- and z-directions of the model with the scoreboard increased by 0.06~0.4 mm and 0.11~0.47 mm, respectively, compared to the model without the scoreboard; under the excitation of the Hollister wave, the peak displacements in the x- and z-directions of the model with the scoreboard increased by 0.05~0.75 mm and 0.08~0.52 mm, respectively; under the excitation of the Cholame wave, the peak displacements in the x- and z-directions of the model with the scoreboard increased by 0.03~0.32 mm and 0.04~0.31 mm, respectively, compared to the model without the scoreboard. In addition, the displacement difference of measuring point D1-0, located in the reticulated shell center between the two models, is the most obvious, as shown in Figure 15a. It is proved that when the center-hung scoreboard is suspended on the suspen-dome structure, the displacement of the suspen-dome structure will increase under the action of earthquakes.

Figure 16 displays the relative error of the peak displacement of each measuring point between the two models. It can be found that the relative error of the peak displacement between the two presents a descending trend from the inner ring to the outer ring; the relative error of the displacement in the z-direction between the two models is taken as an example, when the TAFT wave, Hollister wave, and Cholame wave are input, respectively, the relative errors of the peak displacement of the central measuring point D1-0 between the two models reach 0.48 mm, 0.41 mm, and 0.32 mm, the relative errors of the intermediate measuring point D6-4 between the two models are 0.3 mm, 0.32 mm, and 0.13 mm, while the relative errors of the outermost ring measuring point D48-6 between the two models are only 0.11 mm, 0.09 mm, and 0.04 mm, respectively. It can be inferred that the influence of the center-hung scoreboard on the displacement response of the suspen-dome structure gradually weakens from the inner ring to the structure's outer ring.

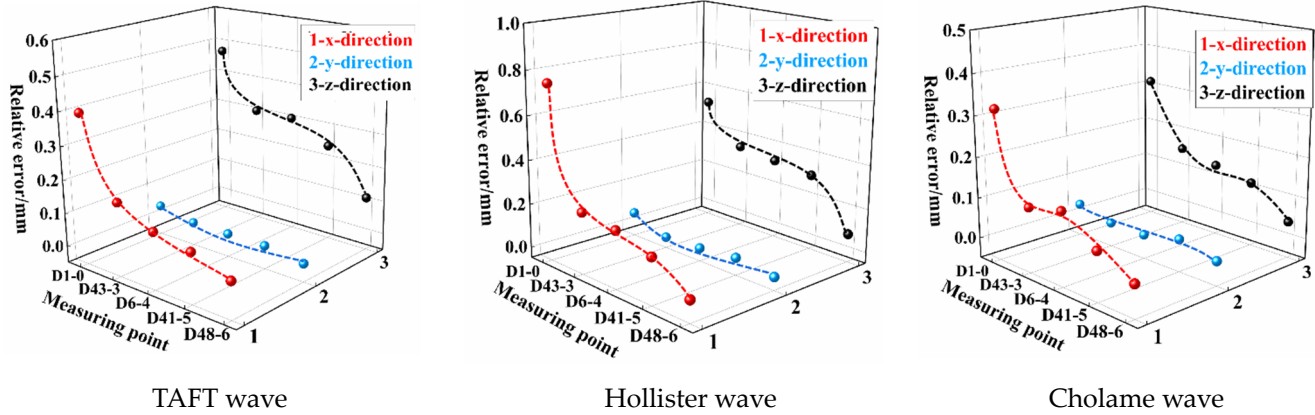

TAFT wave · Hollister wave · Cholame wave

(**a**)

TAFT wave · Hollister wave · Cholame wave

(**b**)

**Figure 15.** Displacement response comparison: (**a**) Peak displacement; (**b**) Displacement time-history curves of measuring point D1-0 in the z-direction.

TAFT wave · Hollister wave · Cholame wave

**Figure 16.** Relative errors of peak displacement between the two models.

### 4.3.2. Displacement Response of Center-Hung Scoreboard

The peak displacement responses of the center-hung scoreboard are listed in Table 10. It can be found that the displacement of the center-hung scoreboard is considerable. For instance, under the action of the Hollister wave, the scoreboard's peak displacements in three directions are 22 mm, 12.24 mm, and 31.05 mm, respectively, far exceeding the displacements of other measuring points. It is shown that during earthquakes, the center-hung scoreboard will swing greatly.

**Table 10.** Peak displacement of center-hung scoreboard.

| Seismic Wave | Peak Displacement/mm | | |
| --- | --- | --- | --- |
| | X-Direction | Y-Direction | Z-Direction |
| Taft wave | 10.71 | 5.98 | 14.02 |
| Hollister wave | 22.00 | 12.24 | 31.05 |
| Cholame wave | 6.58 | 4.31 | 9.03 |

*4.4. Strain Response Comparison of the Models with and without Center-Hung Scoreboard*

Figure 17a shows the peak strain responses of the ring bars, radial bars, and hoop cables, respectively; Figure 17b shows the strain time-history curves of measuring point R45-1. In the test, four strain gauges are arranged outside each member, and the final strain value is the average value of all the strain gauges. It can be seen that the strains of nearly all the measuring points of the ring bars, radial bars, and hoop cables of the model with a center-hung scoreboard are higher than those of the model without a center-hung scoreboard. Under the action of the TAFT wave, the peak strains of the ring bars, radial bars, and hoop cables of the model with the center-hung scoreboard increased by 11.8~30.83 $\mu\varepsilon$, 9.46~50.33 $\mu\varepsilon$, and 19.21~40.28 $\mu\varepsilon$, respectively, compared to the model without the scoreboard; under the action of the Hollister wave, the peak strains of the three types of members of the model with the scoreboard increased by 19.41~54.1 $\mu\varepsilon$, 4.11~49.98 $\mu\varepsilon$, and 15.21~56.88 $\mu\varepsilon$, respectively; under the action of the Cholame wave, the peak strains of the three types of members of the model with the scoreboard increased by 12.0~29.12 $\mu\varepsilon$, 13.03~48.98 $\mu\varepsilon$, and 10.91~18.59 $\mu\varepsilon$, respectively, compared to the model without the scoreboard. In particular, the strain of the bars near the center of the reticulated shell increases most evidently, as shown in Figure 17a. The results demonstrated that when the suspen-dome structure has a center-hung scoreboard, the structural members will have greater internal force under the action of earthquakes, which will increase the deformation of the members.

Figure 18 depicts the relative error of the peak strain of members between the two models. It can be found that the relative error of the peak strain of the three types of members shows a downward trend from the inner ring to the structure's outer ring. Taking the radial bar as an example, when the models are excited by the three seismic waves, the relative error of the peak strain of the central measuring point R45-1 between the two models is as high as about 50 $\mu\varepsilon$, that of the intermediate measuring point R46-4 is all about 25 $\mu\varepsilon$, and that of the outermost ring measuring point R1-7 is all within 12 $\mu\varepsilon$. It can be inferred that the strain of the member closer to the structure's center is more affected by the center-hung scoreboard.

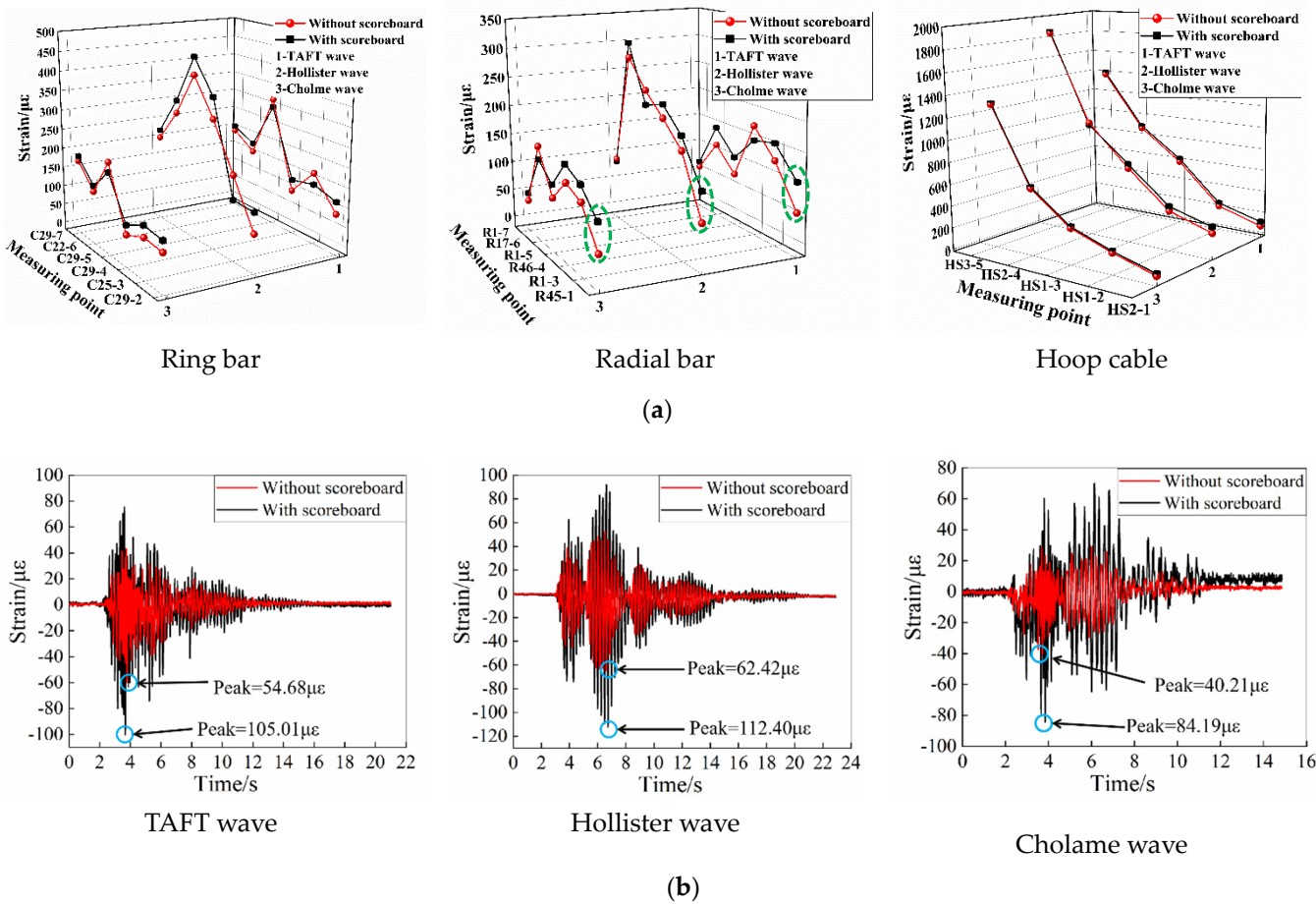

**Figure 17.** Strain response comparison: (**a**) Peak strain; (**b**) Strain time-history curves of measuring point R45-1.

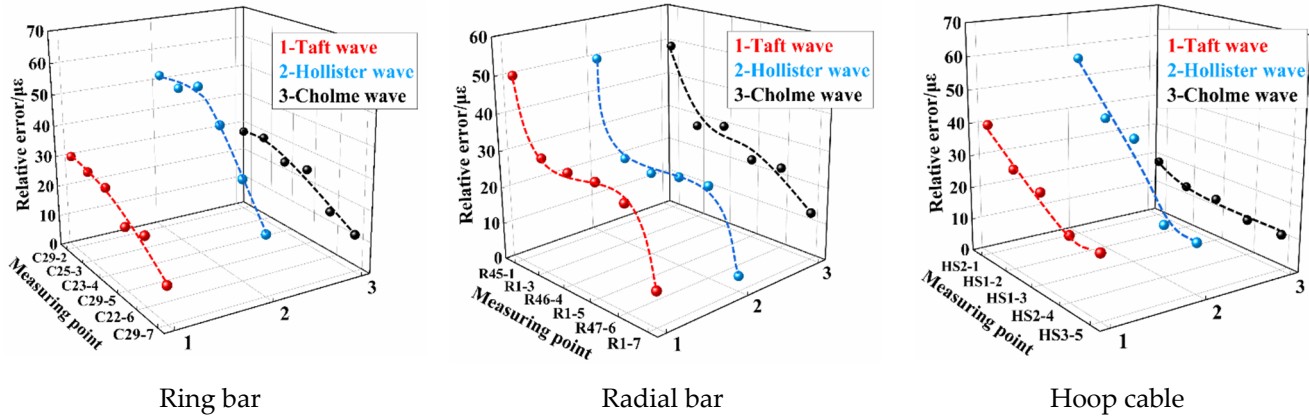

**Figure 18.** Relative errors of peak strain between the two models.

## 5. Discussion and Summary

(1)     According to the comparison of the natural vibration characteristics of the models with and without the center-hung scoreboard, it can be found that the center-hung scoreboard's swing will be excited first as the low-order mode. This is because the scoreboard is flexibly suspended from the roof and has a low swing frequency, so its swing is excited first as a low-order local vibration mode of the suspen-dome structure. Furthermore, the high-order natural frequencies of the suspen-dome structure with the center-hung scoreboard are visibly higher than those of the structure without the

center-hung scoreboard. It is well known that the natural frequency is an important factor in determining the seismic performance of a structure, which is also an essential basis for preventing structural resonance, conducting structural seismic design, and judging structural damage. Therefore, the center-hung scoreboard's influence on the natural vibration characteristics of the suspen-dome structure cannot be ignored in dynamic analysis;

(2) Based on the research results of this paper on the impact of the center-hung scoreboard on the seismic responses of the suspen-dome structure, it can be found that the center-hung scoreboard will increase the seismic response of the suspen-dome structure. For the acceleration response, under the action of different seismic waves, when the scoreboard is suspended on the suspen-dome structure, the maximum acceleration increments in the x-, y- and z-directions of the suspen-dome structure are up to 0.4 g, 0.18 g, and 1.01 g, respectively; meanwhile, under the action of different seismic waves, the maximum acceleration amplification coefficient of the structure with the scoreboard can reach up to 5.62, while that of the structure without the scoreboard is only 4.17. For the displacement response, under the excitation of three seismic waves, the maximum displacement increments in the x- and z-directions of the structure with the scoreboard reach 0.75 mm and 0.52 mm, respectively. For the strain response, under three seismic waves' action, the maximum strain increments of the ring bars, radial bars, and hoop cables of the suspen-dome structure with the scoreboard are as high as 54.19 $\mu\varepsilon$, 50.33 $\mu\varepsilon$, and 56.88 $\mu\varepsilon$, respectively. Consequently, if the influence of the center-hung scoreboard is ignored, the seismic response of the actual suspen-dome structure will be underestimated, and the structure's seismic performance will be overestimated, thereby threatening the structure's safety. It is indicated that the influence of the center-hung scoreboard on the seismic response of the suspen-dome structure should be taken into consideration in seismic analysis;

(3) Combining the influence of the center-hung scoreboard on the measuring points at different positions of the suspen-dome structure, it can be concluded that the influence of the center-hung scoreboard on various seismic responses is gradually weakened from the inner ring to the outer ring of the structure. For the acceleration response, under the action of different seismic waves, the acceleration variation values in the z-direction of the central measuring point A1-0, the intermediate measuring point A4-4, and the outermost ring measuring point A2-6 of the suspen-dome structure with the scoreboard are 0.5~1.0 g, 0.15~0.41 g, and 0.01~0.15 g, respectively, relative to the structure without the scoreboard. For the displacement response, when different seismic waves are input, the displacement variation values in the z-direction of the central measuring point D1-0, the intermediate measuring point D6-4, and the outermost ring measuring point D48-6 of the structure with the scoreboard are 0.32~0.48 mm, 0.13~0.3 mm, and 0.04~0.11 m, respectively, relative to the structure without the scoreboard. For the strain response, under the excitation of different seismic waves, the strain variation values of the central measuring point R45-1, the intermediate measuring point R46-4, and the outermost ring measuring point R1-7 of the structure with the scoreboard are about 50 $\mu\varepsilon$, 25 $\mu\varepsilon$, and 0~12 $\mu\varepsilon$, respectively, relative to the structure without the scoreboard. It can be inferred that when the position of the reticulated shell is closer to the center of gravity of the scoreboard, it will be more seriously affected by the additional inertial effect of the scoreboard, as shown in Figure 19. Therefore its seismic response will increase more obviously. Accordingly, in the seismic design of a suspen-dome structure with a center-hung scoreboard, more attention should be paid to the seismic response of members close to the scoreboard;

(4) From the seismic responses of the center-hung scoreboard under the action of different seismic waves, a conclusion can be drawn that the acceleration response and displacement response of the center-hung scoreboard are tremendous. For the acceleration response, under the action of different seismic waves, the maximum acceleration of

the center-hung scoreboard in three directions can reach 6.15 g, 4.42 g, and 7.41 g, respectively. For the displacement response, under the excitation of different seismic waves, the maximum displacement of the center-hung scoreboard in three directions is up to 22 mm, 12.4 mm, and 31.05 mm, respectively. This is because the scoreboard is only suspended from the reticulated shell by four flexible slings, resulting in it being weakly restrained. Therefore, the center-hung scoreboard is prone to violent movement under dynamic loads and thus has a notable seismic response;

(5) This test mainly studies the effect of the center-hung scoreboard on the natural vibration characteristics and seismic response of the suspen-dome structure. However, compared with the aforementioned shaking table tests of suspen-dome structures [20–22], this test still has certain limitations. Firstly, in the shaking table test, three seismic wave records were selected. More seismic records can be considered to explore more accurately the influence of the center-hung scoreboard on the seismic response of the suspen-dome structure. In addition, this paper mainly analyzes the seismic response of the suspen-dome structure under a single-dimensional earthquake. Therefore, in future experiments or numerical analyses, multi-dimensional seismic excitation can be input to evaluate more comprehensively the seismic performance of the suspen-dome structure with a center-hung scoreboard. Finally, the test results of this paper are only for the suspen-dome structure; whether it is applicable for other types of long-span spatial structures remains to be further verified. Therefore, based on actual engineering, an experimental study on the impact of scoreboards on the seismic performance of other types of long-span spatial structures, such as reticulated shell structures and grid structures, can be carried out;

(6) Most of the measurements in this test are accurate, but some errors still exist. The errors are mainly caused by the following aspects:

    (1) The manufacturing deviation of the test model, including the blanking length of the member, welding between components, etc. In future experiments, more sophisticated machining equipment can be selected, and machining processes can be improved to reduce or eliminate these errors;

    (2) Measurement deviation caused by instruments. During the test, a certain drift inevitably exists in the reading of the strain gauges; this error can be solved by adding more strain gauges and then removing invalid values and averaging them. Furthermore, pull-wire displacement sensors were used to collect the displacement data, so there is inevitable friction between the wire and the instrument. Thus, the effect of this friction can be reduced by applying lubricating oil to the wire;

    (3) Discrepancy between the numerical model and the test model when performing modal analysis of the structure. Although the dead weights of all the members and connecting nodes are considered in the numerical model, the additional weight of the test model (i.e., ear plates, screw bolts, cable head) are difficult to accurately simulate. Moreover, errors caused by measurement and manufacturing deviation can also increase the deviation of numerical and experimental models. In future research, methods to accurately measure the test model's additional mass should be found and more refined numerical models should be built.

(7) In this paper, the effect of the center-hung scoreboard on the natural vibration characteristics and seismic response of suspen-dome structures is studied by means of a shaking table test, and it is proved that the influence is significant and cannot be ignored. The experimental research in this paper is all based on actual engineering, i.e., the Gymnasium of the Lanzhou Olympic Sports Center. Therefore, the research results of this paper will provide help for the installation and use of the center-hung scoreboard in the suspen-dome structure. Furthermore, this paper will contribute to the seismic design of suspen-dome structures with center-hung scoreboards. On the other hand, the design method of this test model can also provide a reference

for designers and researchers to establish dynamic scale models of suspen-dome structures with center-hung scoreboards.

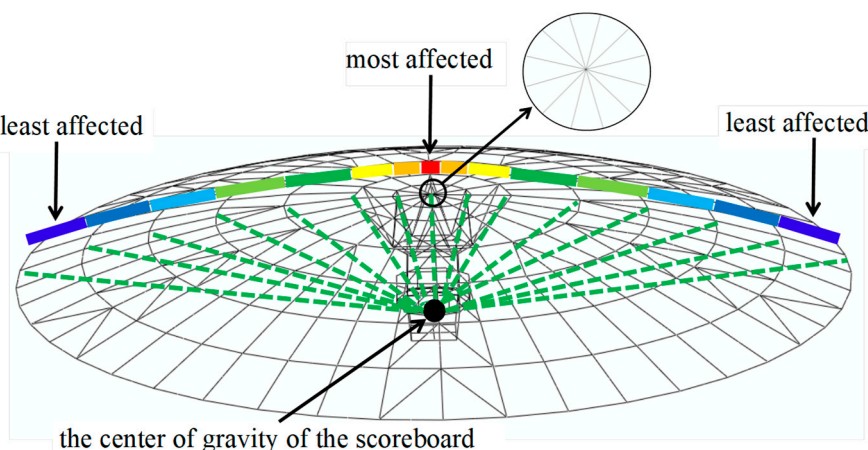

**Figure 19.** The influence of the scoreboard on different positions of the suspen-dome structure.

## 6. Conclusions

In order to investigate the influence of the center-hung scoreboard on the natural vibration characteristics and seismic response of the suspen-dome structure, a scale model was designed and manufactured based on the suspen-dome structure of the Gymnasium of the Lanzhou Olympic Sports Center. Then, in order to more accurately and realistically simulate the dynamic behavior of the structure under the action of an earthquake, shaking table tests were selected and designed. After that, the white noise excitation test was performed on the models with and without the scoreboard, and the natural vibration characteristics of the two models were compared. Moreover, an earthquake simulation test was conducted, and the seismic response of the model with the center-hung scoreboard was compared with the model without the center-hung scoreboard. Finally, the acceleration response and displacement response of the center-hung scoreboard were also analyzed. The following conclusions can be drawn:

(1) The center-hung scoreboard can influence the natural vibration characteristics of the suspen-dome structure. The center-hung scoreboard's swing will be excited first as the low-order mode. Moreover, the center-hung scoreboard will increase the high-order natural frequencies of the overall structure. It is demonstrated that the impact of the center-hung scoreboard on the natural vibration characteristics of the suspen-dome structure should be considered in dynamic analysis;

(2) When the center-hung scoreboard is suspended from the reticulated shell of the suspen-dome structure, the seismic responses of the structure, such as acceleration, displacement, and strain, will increase. In actual engineering, the center-hung scoreboard's influence on the structure's seismic response should not be ignored;

(3) The center-hung scoreboard has a more significant effect on the position closer to its center of gravity; thus, the influence of the scoreboard on various seismic responses of the suspen-dome structure, such as acceleration, displacement, and strain, is gradually weakened from the inner ring to the outer ring of the structure. In seismic design, it is necessary to focus on the seismic response of members near the scoreboard;

(4) Due to being weakly restrained, the center-hung scoreboard's acceleration response and displacement response are tremendous when the suspen-dome structure is excited by seismic loads. In the analysis and design stage, the seismic response of the center-hung scoreboard should be of significant concern;

(5) The research results of this paper have reference value and practical significance for the investigation of the seismic design and seismic performance of suspen-dome structures with center-hung scoreboards. Moreover, the design method of the test

model can also provide a reference for designers and researchers to establish dynamic scale models of suspen-dome structures with center-hung scoreboards.

In future research, designers and researchers can combine the changes in roof load, rise-span ratio, boundary conditions, and other parameters to further study the center-hung scoreboard's influence on the seismic response of the suspen-dome structure. In addition, the impact of the center-hung scoreboard on other seismic performance of the suspen-dome structure, such as the ultimate bearing capacity and failure characteristics of the structure under strong earthquakes, also needs to be investigated.

**Author Contributions:** Conceptualization, S.X.; methodology, S.X., Z.Z. and X.L.; software, Z.Z.; validation, R.L. and T.L.; investigation, Z.Z. and Z.L.; resources, Z.Z.; data Z.Z.; writing—original draft preparation, Z.Z.; writing—review and editing, X.L. and M.D.; visualization, Z.Z. and Z.L.; supervision, X.L. and Q.F.; project administration, X.L. and H.J.; funding acquisition, X.L. All authors have read and agreed to the published version of the manuscript.

**Funding:** This research was funded by National Natural Science Foundation of China (No. 51878014, 51878013, and 51778017. Funder: Xiongyan Li).

**Data Availability Statement:** The raw data required to reproduce these findings cannot be shared at this time due to time limitations. The processed data required to reproduce these findings cannot be shared at this time due to time limitations.

**Acknowledgments:** The authors would like to acknowledge the Lanzhou Olympic Sports Center Construction and Development Co., Ltd., China Aviation Planning and Design Institute (Group) Co., Ltd., Huixian Earthquake Engineering Comprehensive Laboratory of the Institute of Engineering Mechanics, and China Earthquake Administration.

**Conflicts of Interest:** The authors declare no conflict of interest.

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
