# Peer review of "Shaking Table Test Research on the Influence of Center-Hung Scoreboard on Natural Vibration Characteristics and Seismic Response of Suspen-Dome Structures"

_buildings, doi:10.3390/buildings12081231_

Round 1
Reviewer 1 Report
1. The manuscript lacks the proper literature review, the authors shall add a new section “Literature Review”. The literature review section should focus on the following topics:
a. Discuss the system of suspen-dome structures (you may use high-quality drawings for this purpose) and briefly describe the main developments, and improvements of this system over recent history.
b. Discuss the current experimental database limitations in this area.
c. Benefits and limitations of using shake table tests over the other testing tools such as static loading by hydraulic actuators.
2. The structure of the “Introduction” section is unorganized, the authors shall explain the problem statement of this research, motivations, research gap, novelty, and implications of this study.
3. Figure 3b is not clear, the authors should improve the quality of this figure.
4. In figure 4, it is very difficult for the reader to know the location of these pictures in the experimental setup.
5. The section “Analysis of earthquake simulation test results” has low-quality technical writing, the flow and coherence of the writing should be improved. The experimental description and discussion should be expanded and be clearer to describe the experimental results.
6. The section “Discussion” is very short, the authors should address the key findings in this study and explain the results of accelerations, displacements, and strain. The authors should compare these results to other studies in the literature (if exist) and address the key factors that affect these measurements.
7. The limitations and scope of this study should be addressed.
8. The implications and feasibility of this study should be highlighted in the proper location of this manuscript.
9. At the beginning of the section “Conclusions”, you should explain the problem statement, merits, motivations, and approach of this study. The implications and feasibility of this work should be discussed at the appropriate location in the “Conclusions” section.
10. The technical writing in this manuscript needs more improvements, the authors shall improve the writing of this manuscript considering the flow and coherence of discussions.

Author Response
Dear Reviewer and editor,
Thank you for your valuable comments on this article. They have provided significant help for us to improve the article. After careful review, we have responded with the details. Please see the attachment

Reviewer 2 Report
This paper deals with suspen-dome structurewhich is one of the common roof structures of gymnasiums (Gymnasium of Lanzhou Olympic Sports Center). Here a dynamically scaled model with a geometric similarity ratio of 1:20 was established. The white noise excitation test and numerical modal analysis were carried out for the models with and without center-hung scoreboard, respectively, and the influence of the center-hung scoreboard on the natural vibration characteristics of the suspen-dome structure was investigated. Furthermore, various seismic responses of the suspen-dome structure with a center-hung scoreboard were compared with those of the structure without the center-hung scoreboard by conducting earthquake simulation test, the effects of the center-hung scoreboard on the seismic response of the suspen-dome structure are reported. Finally, the acceleration response and displacement response of the center-hung score-board are investigated and analyzed. The research results will provide a reference for further study on the dynamic characteristics and seismic performance of suspen-dome structures with center-hung scoreboards. This paper is well-written and well-organized. It can be published only after addressing the following changes.
1. In the abstract, what was the reason for considering a shaking table test scaled model with a geometric similarity ratio of 1:20? 2. In the abstract, mention the various seismic response studied in this paper. 3. In the introduction, state what are the basic necessary details for a domed indoor stadium and state the primary and secondary tests to be conducted. 4. In Fig 2. (a) and (b) explain the slant bar, ring bar, radial bar, hoop cable, vertical strut, andsteel tie bar with their purpose of the provision in their respective positions. 5. What is the reason for choosing Q355B steel,Q690B steel, and Galfan-coated steel? 6. Fig.5 separate as (a) & (b) and also, and steel construction should be deleted as the entire structure is made out of steel. 7. In page 6 it is mentioned as “The model roof's dead and live loads were 0.8kN/m2 and 0.33kN/m2”: mention the codal recommendations along with the seismic zone. 8. Page 8 section 2.3, more explanation in bullet points is needed for Fig.7 (a)-(g) for better understanding. 9. Page 9. Line 212-213: Explain with reference to table 4 and inference for the same. 10. Section 3.2 numerical simulation using ABAQUS software was conducted. Inside the manuscript, the procedure followed for the Finite Element simulations- must be given separate sub-divisions and mention clearly all the details pertaining to the modelling methodology followed: a) Material properties b) Type of element and mesh sizes c) Load applications and boundary condition d) Material and geometrical imperfection e) Seismic data’s used 11. Three seismic wave records, including the Taft wave, Hollister wave, and Cholame wave were included in this study why not the EL-Centrowave used? 12. What was the peak ground acceleration during the testing while applying the 3 wave patterns in Table 7? -Highlight the inference. 13. Mark the x, y and z directions for the various mode shapes in Table 5, so the discussion part shown in Figs. 12-14 are self-explanatory with the overall suspen-dome structure. 14. Fig. 15 with a zoomed-in figure for the most affected zone would benefit the readers for better understanding. 15. In conclusion point 5 need not be mentioned as one of the conclusions but as a generic statement at the end of conclusion for the future researchers. 16. Please conduct an extensive literature review. Some of the most relevant references which are not in the literature review section are: (I) Performance of a novel slider device in multi-storey cold-formed steel modular buildings under seismic loading (Ii) Seismic protection of modular buildings with bonded rubber unit sliders: Experimental study (iii) Seismic protection of modular buildings with galvanised steel wall and bonded rubber units: Experimental and numerical study
Author Response
Dear Reviewer and editor,
Thank you for your valuable comments on this article. They have provided significant help for us to improve the article. After careful review, we have responded with the details. Please see the attachment.

Round 2
Reviewer 1 Report
The manuscript has been improved, but the authors should address the following two comments:
1) The section "Literature review" should be located after the section "Introduction". I mean Section 1 is "Literature" and Section 2 is "Introduction".
2) There are some typos in the manuscript, the authors should perform a comprehensive check to avoid typos.
Author Response
Dear Manuscript Reviewers and Editors,
Thank you for your valuable comments and continued interest section of the article. Based on your comments, we have revised:
- After the "Literary Criticism" section as "Introduction". The specific content is "1 Introduction: 1.1 Literature Review; 1.2 Introduction".
- 2. Corrected typos in the updated manuscript.
Reviewer 2 Report
The authors have addressed my comments. The paper can be accepted for publication.
Author Response
Dear Reviewer and Editor,
Thank you for your valuable guidance and recognition of this article.